# Analysis of the Toll and Spaetzle Genes Involved in Toll Pathway-Dependent Antimicrobial Gene Induction in the Red Flour Beetle, *Tribolium castaneum* (Coleoptera; Tenebrionidae)

**DOI:** 10.3390/ijms24021523

**Published:** 2023-01-12

**Authors:** Daiki Kato, Ken Miura, Kakeru Yokoi

**Affiliations:** 1Applied Entomology Laboratory, Graduate School of Bioagricultural Sciences, Nagoya University, Furo-cho, Chikusa, Nagoya 464-8601, Japan; 2Insect Design Technology Group, Division of Insect Advanced Technology, Institute of Agrobiological Sciences, National Agriculture and Food Research Organization (NARO), 1-2 Owashi, Tsukuba 305-8519, Japan

**Keywords:** Toll, spaetzle, *Tribolium castaneum*, Toll pathway

## Abstract

Insects rely only on their innate immune system to protect themselves from pathogens. Antimicrobial peptide (AMP) production is the main immune reaction in insects. In *Drosophila melanogaster*, the reaction is regulated mainly by the Toll and immune deficiency (IMD) pathways. Spaetzle proteins, activated by immune signals from upstream components, bind to Toll proteins, thus, activating the Toll pathway, which in turn, induces AMP genes. Previous studies have shown the difference in immune systems related to Toll and IMD pathways between *D. melanogaster* and *Tribolium castaneum*. In *T. castaneum*, nine Toll and seven spaetzle (spz) genes were identified. To extend our understanding of AMP production by *T. castaneum*, we conducted functional assays of Toll and spaetzle genes related to Toll-pathway-dependent AMP gene expression in *T. castaneum* under challenge with bacteria or budding yeast. The results revealed that *Toll3* and *Toll4* double-knockdown and *spz7* knockdown strongly and moderately reduced the Toll-pathway-dependent expression of AMP genes, respectively. Moreover, *Toll3* and *Toll4* double-knockdown pupae more rapidly succumbed to entomopathogenic bacteria than the control pupae, but *spz7* knockdown pupae did not. The results suggest that *Toll3* and *Toll4* play a large role in Toll-pathway-dependent immune reactions, whereas *spz7* plays a small part.

## 1. Introduction

Insects possess only innate immune systems, which are divided into cellular immunity and humoral immunity [1]. One of the main aspects of humoral immunity is antimicrobial peptide (AMP) production. The mechanisms of AMP production at genetic and molecular levels were investigated in *Drosophila melanogaster* (Diptera; Drosophilidae, Meigan, 1830) as a model insect [2]. The induction of AMP genes by microbial infection is regulated by two signaling pathways, namely, the Toll and immune deficiency (IMD) pathways [3,4]. In *D. melanogaster*, fungi, or yeasts are recognized by Gram-negative binding protein (GNBP) 3, whereas Gram-positive bacteria are recognized by GNBP1 and peptidoglycan recognition protein (PGRP)-SA. Through this recognition, immune signals are triggered [5,6,7]. After the signals are transduced via the serine protease cascade, spaetzle processing enzyme (SPE), which is located at the end of the serine protease cascade, is activated to cleave pro-spaetzle into activated spaetzle (spz) [8,9]. The cleaved spz binds to Toll proteins (transmembrane receptors) and activates the Toll pathway [10,11]. Consequently, Dorsal or Dif (NF-κB transcription factors), which is located at the end of the Toll pathway, is activated and induces the expression of Toll-pathway-dependent AMP genes (e.g., *drosomycin*) [12]. Gram-negative bacteria are mainly recognized by PGRP-LC and LE, leading to the activation of the IMD pathway. The immune signals stimulated by this recognition are transduced through intracellular components of the IMD pathway, such as IMD, DREDD, and IAP2. The signals activate Relish, a NF-κB transcription factor located at the end of the IMD pathway, which in turn, induces several AMP genes (e.g., *diptericin*) [13].

In our previous study, to compare AMP gene induction systems between *D. melanogaster* and non-*Drosophila* species, we investigated the AMP gene induction system of the red flour beetle *Tribolium castaneum* (Coleoptera; Tenebrionidae, Herbst, 1797) using *T. castaneum* genome data and RNA interference (RNAi), which has good knockdown effect against *T. castaneum*. Our results revealed that 18 AMPs of *T. castaneum* were classified into four types according to the gene induction profiles by microbes [14]. Furthermore, whether the four types of AMP were regulated by the Toll or IMD pathway or both were determined using *IMD* or *MyD88* (representative genes of the IMD and Toll pathways, respectively) knockdown pupae. The results showed that the AMPs were classified into IMD-pathway-dependent AMP, Toll-pathway-dependent AMP, AMP dependent of both pathways, or AMP independent of either pathway. *Cecropin2* (*Cec2*) and *Cec3* are Toll-dependent AMPs with slow and sustained induction that was moderate or weak compared to IMD-dependent AMPs or AMPs dependent of both IMD and Toll pathways [15]. Furthermore, to elucidate the mechanism of the Toll and IMD pathways in *T. castaneum,* we determined whether orthologues of Toll and IMD pathway genes were involved in Toll- or IMD-dependent signaling. The results showed that canonical signal transductions via intracellular Toll and IMD pathway components were conserved, and both pathways were activated by both Gram-negative and -positive bacteria and yeast because PGRP-LA was mainly involved in AMP induction by both types of bacteria, which was the most distinctive difference from the AMP production system of *D. melanogaster* [16,17,18].

The molecular mechanisms of Toll and spz interactions during immune signaling in *D. melanogaster* are well understood, and they play key roles in Toll-pathway-mediated immune reactions. Toll proteins possess an ectodomain mainly composed of leucine-rich repeats and a cytosolic homology domain called the Toll/IL-1R (TIR) domain, which interacts with adaptor molecules [19,20]. Spz proteins are normally inactivated owing to the masking of the predominantly hydrophobic C-terminal spz region [8,21,22]. When immune signals are transduced, spz proteins are proteolytically cleaved by SPE and activated [9]. The activated C-terminal region of spz formed cystine knot motifs and dimers, which act as ligands of Toll. Complexes of spz and Toll proteins arrange hetero-trimers consisting of MyD88, Tube, and Pelle, which are intracellular components of the Toll pathway, leading to the expression of Toll-pathway-dependent AMPs (e.g., drosomycin) [10,11,23,24]. These involvements of Toll and spz in AMP production were reported in *Bombyx mori* (Lepidoptera; Bombycidae, Linnaeus, 1758)*, Tenebrio molitor* (Coleoptera; Tenebrionidae, Linnaeus, 1758), and *Manduca sexta* (Lepidoptera; Sphingidae, Linnaeus, 1763) [25,26,27,28,29].

In *T. castaneum*, nine Toll-related genes were identified. Four of ten Toll genes form a clade next to *D. melanogaster Toll1* (Tl) and are involved in Toll-dependent AMP production. Moreover, seven spz genes were identified in *T. castaneum* [30]. 

In the present study, to expand our understanding of the immune system related to the Toll pathway in *T. castaneum*, we assessed the function of the Toll and spz genes of *T. castaneum* in AMP production. We first performed sequencing analysis, which is the phylogenetic analysis and sequence aliments (only spz genes). Changes in the gene expression of *T. castaneum* pupae after microbe challenges were measured. Furthermore, the Toll or spz genes involved in Toll-pathway-dependent AMP production due to microbe challenge were determined using RNAi and quantitative reverse transcript PCR (qRT-PCR). Finally, we tested whether the Toll and spz genes were involved in immune reactions against two entomopathogenic bacteria.

## 2. Results

### 2.1. Functional Assay of Toll Genes

#### 2.1.1. Phylogenetic Analysis of Toll Genes

To screen for functional *T. castaneum* Toll genes, phylogenetic analysis was performed using amino acid sequences of Toll genes in *T. castaneum*, *D. melanogaster*, and *Aedes aegypti* (Diptera: Culicidae, Linnaeus in Hasselquist, 1762). As shown in Figure 1, *T. castaneum* Toll1−4 (TcToll1−4) formed a clade next to the clade including DmToll1, AaToll1A, and Aa Toll1B, which is consistent with a previous finding [30]. These results suggest that Tc-Toll1−4 is involved in Toll-pathway-dependent AMP production, similar to DmToll1. Therefore, Toll1−4 were selected for further analysis.

#### 2.1.2. Toll Gene Induction by Microbial Challenge

Several immune-related genes, such as AMP genes, are induced by microbial infection. Therefore, we examined whether *Toll1−4* genes were induced by heat-killed *Escherichia coli* (Gram-negative bacteria; Ec), *Micrococcus luteus* (Gram-negative bacteria; Ml), and *Saccharomyces cerevisiae* (Eukaryote; Sc) at 6 and 24 h after injection of the microbes (the expression of AMP genes peaked at 6 or 24 h post-injection) (Figure 2). Only Toll3 was significantly induced by Ec and Sc at 6 h post-injection (Figure 2C).

#### 2.1.3. Determination of Whether *Toll1−4* Are Involved in Toll-Pathway-Dependent AMP Gene Expression

To determine whether *Toll1−4* contribute to Toll-pathway-dependent AMP gene (*Cec2* and *Cec3*) expression, pupae with single or multiple knockdown of *Toll1−4* were prepared by RNAi. In the pupae injected with one or two Toll gene double-stranded RNAs (dsRNAs), the *Cec2* and *Cec3* mRNA levels were significantly lower than those in the control (injected with maltose binding protein E (*malE*) dsRNA) (Appendix A).

Pupae with *Toll1, Toll2, Toll3*, or *Toll4* knockdown were challenged with either heat-killed Ec, Ml, or Sc to elucidate the role of the four Toll genes in *Cec2* induction (Figure 3). *Cec2* mRNA levels in *Toll4* knockdown pupae at 6 and 24 h post-challenge with Ec were significantly lower than those in the control (*malE* dsRNA-treated pupae); moreover, the decrease in *Cec2* expression in *Toll4* knockdown pupae at 24 h post-challenge was greater than that at 6 h post-challenge (Figure 3A). However, the expression levels of *Cec2 i*n other Toll knockdown pupae were not significantly lower than those in the control. After the Ml challenge, *Cec2* mRNA levels in Toll4 knockdown pupae at 24 h post-challenge were significantly lower than those in the control, whereas those in the other knockdown pupae or *Toll4* knockdown pupae at 6 h post-challenge were not significantly different from those in the control (Figure 3B). After the Sc challenge, *Cec2* mRNA levels in either *Toll1−4* knockdown pupae at 6 or 24 h post-challenge were not significantly lower than those in the control (Figure 3C).

Moreover, the effect of *Toll1−4* knockdown on *Cec3* induction was investigated (Figure 4). *Cec3* mRNA levels in Toll4 knockdown pupae at 6 or 24 h post-challenge with Ec were significantly lower than those in the control, and the decrease in *Cec3* expression at 24 h post-challenge was greater than that at 6 h post-challenge, similar to the finding on *Cec2* expression (Figure 4A). The induction of *Cec3* at 6 and 24 h after Ml challenge was significantly altered by either *Toll1* or *Toll4* knockdown (Figure 4B); the greatest decrease in Cec3 expression was observed in Toll4 knockdown pupae at 24 h post-challenge. After the Sc challenge, a significant reduction in *Cec3* expression was observed in *Toll4* knockdown pupae at 24 h post-challenge (Figure 4C). These results suggest that Toll4 is involved in Toll-dependent AMP gene expression at 24 h post-challenge with Ec or Ml.

As described in the introduction, the Toll protein forms dimers when immune signals are transduced. Therefore, we examined whether Toll genes acted redundantly or synergistically in Toll-pathway-dependent AMP gene expression. For this purpose, we prepared pupae in which two out of the four Toll genes were knocked down. First, we examined whether the double-knockdown of Toll genes affected *Cec2* induction by Ec, Ml, or Sc challenge (Figure 5). After the Ec challenge, the mRNA levels of *Cec2* in Toll3+Toll4 double-knockdown pupae at 6 and 24 h post-challenge were strongly decreased compared with those in the control pupae; in contrast, *Cec2* mRNA levels in *Toll3+Toll4* double-knockdown pupae at 24 h post-challenge were only moderately lower than those in the control (Figure 5A). After the Ml challenge, the expression levels of *Cec2* in *Toll3+Toll4* double-knockdown pupae at 6 and 24 h post-challenge were much lower than those in the control; on the contrary, *Cec2* expression levels in Toll1+Toll4 or Toll3+Toll4 knockdown pupae at 24 h post-challenge were only moderately lower than those in the control (Figure 5B). After the Sc challenge, the mRNA levels of *Cec2* in *Toll3+Toll4* double-knockdown pupae at 6 and 24 h post-challenge were much lower than those in the control (Figure 5C); in addition, the *Cec2* level in *Toll2+Toll4* double-knockdown pupae at 24 h post-challenge was dramatically lower than that in the control, whereas that in either *Toll1+Toll2*, *Toll1+Toll3*, or *Toll1+Toll4* double-knockdown pupae was mildly but significantly lower than that in the control. Second, we investigated the effect of double-knockdown of two Toll genes on *Cec3* induction (Figure 6). After Ec challenge, over 50% reductions in *Cec3* mRNA level were observed in *Toll3+Toll4* double-knockdown pupae at 6 and 24 h post-challenge compared to that in the controls, and approximately 20–30% reductions were observed in *Toll2+Toll4* double-knockdown pupae at 6 and 24 h post-challenge compared to that in the control (Figure 6A). After Ml challenge, over 70% reductions in *Cec3* expression levels were observed in *Toll3+Toll4* double-knockdown pupae at 6 and 24 h post-challenge compared to that in the control, and approximately 50% reductions were observed in either *Toll1+Toll4* or *Toll2+Toll4* double-knockdown pupae at 24 h post-challenge compared to that in the control (Figure 6B). After the Sc challenge, approximately over 70% reductions in *Cec3* mRNA level were observed in *Toll3+Toll4* double-knockdown pupae at 6 and 24 h post-challenge and in *Toll2+Toll4* double-knockdown pupae at 24 h post-challenge compared to those in the control, and approximately 20–40% reductions were observed in either *Toll1+Toll2, Toll1+Toll3*, or *Toll1+Toll4* double-knockdown pupae compared to those in the control (Figure 6C). These results suggest that *Toll3* and *Toll4* redundantly play central roles in the induction of *Cec2* or *Cec3* by microbial challenge.

#### 2.1.4. Survival Assay to Determine the Role of *Toll3* and *Toll4* in the Immune Response against the Two Entomopathogenic Bacteria

As described in the previous section, *Toll3* and *Toll4* were involved in the induction of *Cec2* and *Cec3* (Toll-pathway-dependent AMP genes). To investigate whether these genes participate in the immune response, *Toll3* and *Toll4* double-knockdown pupae, malE dsRNA-treated pupae (control), and *IMD* knockdown pupae (positive control) were injected with two entomopathogenic bacteria, *Enterobacter cloacae* (Gram-negative bacteria with meso-diaminopimelic acid (DAP)-type peptidoglycan) and *Bacillus subtilis* (Gram-positive bacteria with DAP-type peptidoglycan). The number of living pupae was counted every 24 h. After *E. cloacae* challenge, no significant differences in survival curves were observed between the control and *Toll3*+*Toll4* double-knockdown pupae, whereas the survival curve of *IMD* knockdown pupae (positive control) was significantly different from that of the control (Figure 7A). After *B. subtilis* challenge, the number of dead pupae was significantly higher in the *Toll3+Toll4* double-knockdown group than in the control group. However, *p* value of control vs. *Toll3* and *Toll4* knockdown pupae were not lower than that of control vs. *IMD* knockdown, suggesting that *Toll3* and *Toll4* contributed to immune defenses against *B. subtilis*, but the degree of *Toll3* and *Toll4* contributions to the defenses was not the same as that of the *IMD* contributions.

### 2.2. Functional Assay of spz Genes

#### 2.2.1. Phylogenetic and Sequence Analyses of the spz Gene

To identify the *T. castaneum* spz genes (Tc-spz) involved in Toll-pathway-dependent AMP gene expression, we performed phylogenetic analysis using spz sequences of various insect species, *D. melanogaster* (Dm-spz), *Anopheles gambiae* (Diptera; Culicidae, Giles, 1902) (Ag-spz), *A. aegypti* (Aa-spz), and *M. sexta* (Ms-spz) (Appendix A), because spz sequences are not similar to each other. The phylogenetic tree constructed showed that spz1 genes of the used species were scattered in the tree (Figure 8A). Furthermore, the sequence analysis of C-terminal 106 amino acid residues of the spz protein, which plays key roles for functioning as the signal transducers, was performed [21]. One hundred twenty amino acid sequences of Tc-spz1-7 and spz1 of the non-*T. castaneum* species from C-terminal (Appendix A) were aligned to search the arginine residues and seven cysteine residues, which are predicted proteolytic cleavage sites and are predicted to formthe cysteine knot, respectively (Figure 8B) [21,27]. The results showed that Tc-spz6 did not possess the arginine residues of the proteolytic cleavage site, and Tc-spz2, 4, 6, and 7 did not completely possess the seven cysteine residues required for the cysteine knot. In conclusion, from both in silico analyses, we did not determine that the candidate *T. castaneum* spz genes were involved in the immune reactions.

#### 2.2.2. spz Gene Induction by Microbial Challenge

We also examined whether spz gene expression is induced by either Ec, Ml, or Sc challenges (Figure 9). Although *spz4* mRNA amounts in pupae Ec or Sc post 24 h challenge were lower than Uc controls (Figure 9D), the expression levels of *spz7* in pupae Ml or Sc at 6 h post-challenge and phosphate-buffered saline (PBS), Ec, Ml, or Sc at 24 h post challenge were significantly higher than those of *spz7* in Uc controls. These results suggested that *spz7* can be a candidate gene involved in immune reactions against the microbes (Figure 9G).

#### 2.2.3. Determination of spz Genes Involved in Toll-Pathway-Dependent AMP Gene Expression

We determined whether spz genes were involved in *Cec2* and *Cec3* induction, as the candidate spz genes were not determined from sequencing analysis results. The mRNA levels of *Cec2* and *Cec3* in all spz knockdown pupae, in which the expression of all seven spz genes were significantly reduced, were compared with those in the control (Appendix A) at 24 h post-challenge with Ec, Ml, and Sc (Figure 10). This was based on previous reports that *Cec2* and *Cec3* induction peaked at 24 h post-challenge [15]. *Cec3* mRNA levels in all *spz* knockdown pupae were lower than those in the control, and both *Cec2* and *Cec3* mRNA levels in all *spz* knockdown pupae were significantly lower than those in the control, suggesting that spz genes play a role in the Toll-pathway-dependent induction of AMP genes (*Cec2* and *Cec3*).

To determine the spz gene involved in the induction of AMP genes by Ml, we generated pupae in which six *spz* genes were knocked down (e.g., spz(All-3) indicates knockdown of *spz1, 2, 4, 5, 6,* and *7*), and the pupae were then challenged with Ml. At 24 h post-challenge, the mRNA levels of *Cec2* and *Cec3* in the six spz gene knockdown pupae and the control were compared (Figure 11).

At 24 h post-Ml challenge, the level of *Cec2* mRNA in the pupae possessing only the spz7 gene was significantly different from that in the pupae with knockdown of all spz genes but not different from that in the control pupae. In contrast, the mRNA level of *Cec2* in the pupae possessing only the spz1, 2, 3, 4, 5, or 6 gene was significantly different from that in the control pupae, but not different from that in the pupae with knockdown of all spz genes (Figure 11A,B). The level of *Cec3* mRNA in the pupae possessing only the spz7 gene was significantly different from that in the pupae with knockdown of all spz genes and the control pupae. Moreover, *Cec3* mRNA level in the pupae possessing only the spz1, 2, 3, 4, 5, or 6 gene was significantly different from that in the control pupae but not different from that in the pupae with knockdown of all spz genes (Figure 11C,D). These results indicate that spz7 is involved in the *Cec2* and *Cec3* gene induction.

To further confirm this finding, *Cec2* and *Cec3* mRNA levels in spz7 knockdown pupae and control pupae at 24 h post-challenge with Ec, Ml, or Sc were compared (Figure 12). The results showed a significant reduction of approximately 65% in *Cec2* mRNA level in spz7 knockdown pupae challenged with Ml compared with the control pupae. A significant reduction of 50% in *Cec3* mRNA levels induced by Ml challenge was also observed in spz7 knockdown pupae compared to those in control pupae (Figure 12A,B). A significant and relatively moderate reduction (compared to *Cec2* mRNA reduction.) in *Cec3* mRNA level was observed in *spz7* knockdown pupae challenged with Sc compared with the control pupae (Figure 12B). These results suggest that *spz7* plays a role in the induction of Toll-pathway-dependent AMP genes by Ml.

#### 2.2.4. Survival Assay to Determine the Role of *spz7* in the Immune Response against Entomopathogenic Bacteria

In the previous experiment, *spz7* was shown to be involved in Cec2 and Cec3 induction by Ml. For further functional analysis of *spz7,* a survival assay using *spz7* knockdown pupae was conducted as described in Section 2.1.4 (Figure 13). There was no difference in survival curves between control and *spz7* knockdown pupae infected with *E. cloacae* (Figure 13A). In contrast, *spz7* knockdown pupae infected with *B. subtilis* died earlier than the control pupae, but the difference was not significant (Figure 13B).

## 3. Discussion

In this study, to enhance our understanding of the mechanisms of immune responses in *T. castaneum*, especially AMP production mediated by the Toll pathway, we determined the Toll and spz genes involved in the induction of *Cec2* and *Cec3*, which are Toll-pathway-dependent AMP genes. Through phylogenetic analysis, four candidate *Toll* genes were screened from seven Toll genes. Gene expression analysis showed that *Toll3* and *spz7* were induced by challenge with Ec, Ml, or Sc, suggesting their involvement in the immune function. Experiments in pupae subjected to RNAi revealed that *Toll4* was involved in *Cec2* and *Cec3* induction by Ec, Ml, or Sc, whereas *Toll3* acted synergistically with *Toll4* in immune reactions against *B. subtilis*. In addition, *spz7* was involved in *Cec2* and *Cec3* induction by Ml only, and the reduction in *Cec2* and *Cec3* expression was weaker in spz7 knockdown pupae than in *Toll3* and *Toll4* double-knockdown pupae, compared to that in the control. Furthermore, a survival assay showed that *spz7* knockdown pupae succumbed to *B. subtilis* earlier than the control, but the difference was not significant.

Although *Toll4* expression was not induced by microbial challenge, *Toll4* knockdown significantly altered *Cec2* and *Cec3* expression levels at 24 h post-challenge with Ec, Ml, or Sc. Moreover, at 6 h post-challenge, Toll knockdown significantly reduced *Cec2* induction by Ec and *Cec3* induction by Ec and Ml. The reduction at 24 h post-challenge was approximately 30–70% to the control level. In addition, *Toll3* and *Toll4* double-knockdown severely reduced *Cec2* and *Cec3* induction at 6 and 24 h post-challenge with Ec, Ml, or Sc. These results suggest that *Toll3* and *Toll4* are mainly involved in the expression of Toll pathway-dependent AMP genes and that the two genes work redundantly. Structural analysis revealed that Toll proteins formed dimers, which in turn, function as signal transducers of Toll pathway signals [10,11,24]. We presumed that in *T. castaneum, Toll3* and *Toll4,* homodimers redundantly participated in the signal transduction of Toll-pathway immune signals, leading to *Cec2* and *Cec3* induction, with *Toll4* homodimers having a greater contribution than *Toll3* homodimers. The survival assay showed that *Toll3* and *Toll4* double-knockdown pupae succumbed earlier to *B. subtilis,* but the degree was not as severe as that in IMD knockdown pupae. *Dif1* and *Dif2* double-knockdown pupae also succumbed earlier to *B. subtilis,* showing the same degree as that in *Toll3* and *Toll4* knockdown pupae, but this phenomenon was not observed in insects with knockdown of *MyD88, Tule,* or *Pelle*, which are intracellular components of the Toll pathway [15,16]. The reason may be the crosstalk between the intracellular components of the IMD and Toll pathways, which was reported in *D. melanogaster* and other insects [31,32].

Among the seven spz genes of *T. castaneum*, only *spz7* was involved in *Cec2* and *Cec3* induction at 24 h after Ml challenge. Phylogenetic analysis showed that *spz7* (Tc-spz7) was located in a clade next to Dm-spz1, which is related to Toll-pathway-dependent AMP gene expression [8]. However, the bootstrap values of the spz phylogenetic tree were lower than those of the Toll tree, suggesting that the sequences of spz genes varied. Sequence alignment results showed that the *spz7* protein did not possess the cysteine residue used for the dimerization of spz proteins. A structural report of *D. melanogaster* spz showed that the forming cysteine-knot structure and dimerization of spz protein were indispensable for the interactions with Toll proteins to transduce immune signaling [21,22]. Considering these findings, we presumed that *T. castaneum* spz could function without dimerization, or there is a nucleotide sequence error in *spz7*, as the correct sequence should contain the cysteine residue required for dimerization.

As described above, *Toll3* and *Toll4* double-knockdown dramatically reduced *Cec2* and *Cec3* induction (over 90% reduction compared to that in the controls) at 24 h after microbial challenge (Ec, Ml, and Sc). Toll3 and Toll4 were shown to contribute to immune reactions against the two entomopathogenic bacteria to some extent. In addition, *spz7* knockdown moderately reduced *Cec2* and *Cec3* expression (at most approximately 60% compared to that in the controls), which was induced by Ml only. Furthermore, the survival assay showed that spz7 did not contribute to the immune reaction against the two entomopathogenic bacteria. The *Cec2* and *Cec3* expression reduction ratio was different between Toll and spz genes. We presumed two hypotheses. One is that Toll functions as a direct recognition protein, such as the Toll-like receptor (TLR) protein [33]. TLR proteins in mammals directly bind to pathogen-derived structures, such as Gram-negative bacterial lipopolysaccharides, and viral DNA, and activate the innate immune reactions. Therefore, *T. castaneum* dimers consisting of *Toll3* and *Toll4* might function as a TLR while canonical immune signals aroused from PGRP and GNBP are transduced via spz and Toll complexes. The functional analysis of immune genes upstream of Toll and spz, such as PGRP-SA or serine proteases, could be a key for determining whether the hypothesis is true. If the upstream genes, in fact, do not function, the “Toll direct-recognition system” will function. The other hypothesis is that there are unidentified *T. castaneum* spz genes acting as a spz gene in *D. melanogaster*. As revealed by the sequence analysis of spz proteins in the present study, spz sequences might vary across species. In previous reports, seven spz genes were identified in the *T. castaneum* genome [30]. However, we presumed that there might be more spz genes in the *T. castaneum* genome. Owing to advances in sequencing devices, more accurate genome data can be constructed [34]. For example, the first genome sequence data of *Apis mellifera* (Hymenoptera; Apidae, Linnaeus, 1758) were published in 2006, and after several updates, the latest version of the *A. mellifera* genome is at chromosome level with very high continuity [35,36,37]. In comparison, the gene set data used in this study were released in 2007, and the latest version of the *T. castaneum* gene set was released in 2020 [30,38,39]. However, using only 2007 version gene set data, the gene set analysis focusing on immune-related genes was performed [30]. Screening and functional analysis of new spz genes in the new gene set can be conducted in future research to improve our knowledge of the immune system of *T. castaneum*.

In conclusion, we revealed that among nine Toll genes and seven spz genes, *Toll3*, *Toll4*, and *spz7* were involved in Toll-dependent AMP gene induction by microbial challenge. Our findings also demonstrated differences in their degree of contribution to *Cec2* and *Cec3* induction and immune reactions against entomopathogenic bacteria. These findings improve our understanding of the immune system of non-*Drosophila* insect species and provide a new possible mechanism of AMP production reaction in insects.

## 4. Materials and Methods

### 4.1. Insect Rearing

*T. castaneum* was reared on whole wheat flour at 30 °C in the dark. For use in experiments, the pupae were staged as described in [14].

### 4.2. Microbes and Injections

Heat-killed *E. coli* DH5a, *M. luteus* ATCC4698, and *S. cerevisiae* S288C, which are representative microbes of Gram-negative bacteria, Gram-positive bacteria, and eukaryotes, respectively, were prepared as described in our previous paper [15]. Fifty nanoliters of Ec, Ml, or Sc suspension equivalent to 2.9 × 10^8^, 2.9 × 10^7^, and 6.3 × 10^6^ cells/mL in PBS, respectively, was injected using Nanoject II (Drummond Scientific Company, Broomall, PA, USA) into day-3 pupae pretreated with dsRNA or naïve day-3 pupae. For the survival assay, living *E. cloacae* and *B. subtilis* were prepared as described in our previous paper [14]. *S. cerevisiae S288C* was kindly gifted from Dr. T. Ushimaru of Shizuoka University. *M. luteus* ATCC4698 was supplied by the RIKEN Bioresource Center in Japan.

### 4.3. Gene Sequence Analysis

*T. castaneum* gene sequences used in the present study were retrieved from glean_tribolium.dna.fa and glean_tribolium.protein.fa compressed as TCGleanPrediction.tar.gz located in Tcas2.0/annotation/TCGleanPrediction data of reference genome version Tcas2.0 (able to access via FTP server, URL: ftp://ftp.hgsc.bcm.edu/Tcastaneum/ or “BCM-HGSC data” in the “Red Flour Beetle Genome Project” site (URL: https://www.hgsc.bcm.edu/arthropods/red-flour-beetle-genome-project) accessed on 10 January 2023) [30,38]. The accession IDs of the gene sequences used in the study are listed in Appendix A. Non-*T. castaneum* gene sequences were also retrieved from public databases using the accession IDs shown in Appendix A. Sequence alignments and phylogenetic tree construction were performed using ClustalW version 2.1 in the DNA Data Bank of Japan server (URL: https://www.ddbj.nig.ac.jp/services/clustalw-e.html; temporally halted at 27 December 2022, accessed on 10 January 2023) [40]. Alignment results of C-terminal spz were visualized using BOXSHADE with default settings (URL: http://arete.ibb.waw.pl/PL/html/boxshade.html, accessed on 10 January 2023) [41].

### 4.4. RNA Extraction and qRT-PCR

The total RNA extraction method was described in [14,15]. Briefly, 0.5 μg of quality checked total RNA was converted into first-strand cDNA using a PrimeScript RT Reagent Kit with gDNA Eraser (TaKaRa Bio Inc., Shiga, Japan), according to manufacturer’s instructions. qRT-PCR was performed using first-strand cDNA and a primer pair designed for target genes using the SYBR Premix Ex Taq Perfect Real Time Kit Tli RNAaseH Plus (TAKARA) and Thermal Cycler Dice Real Time System (Model TP800; TaKaRa Bio Inc.). The sequences of the primers for qRT-PCR are shown in Appendix A. The qRT-PCR conditions and the calculation method of mRNA levels of target genes are described in [14].

### 4.5. RNAi

RNAi was performed as described in our previous study [15]. Briefly, cDNA with T7 RNA polymerase promoter sequence, which was generated by conventional PCR, was used as a template for dsRNA synthesis using a MEGAscript RNAi Kit (Ambion, Waltham, MA, USA). Next, 100 ng of dsRNA for each target gene was used for experiments. The dsRNA was injected using Nanoject II into day-0 pupae, followed by incubation for 72 h. The incubated pupae were then examined for knockdown efficiency or challenged with microbes. dsRNA with a partial maltose binding protein E (malE) sequence was used as a negative control. The sequences of primer pairs used in the synthesis of cDNA template are shown in Appendix A.

### 4.6. Survival Assay

Survival assays were performed as described in our previous paper [15]. The incubated pupae pretreated with dsRNA were injected with live *E. cloacae* (50 nl, A600 = 0.1) or *B. subtilis* (100 nl, A600 = 3.0). The number of surviving pupae was counted every 24 h. Data are shown in Kaplan–Meier plots, and statistical analysis (the Gehan–Breslow–Wilcoxon test) was performed using a software package, Ekuseru-Toukei 2010 (Social Survey Research Information Co., Ltd., Tokyo, Japan). *E. cloacae* and *B. subtilis* were generous gifts from Dr. Y. Yagi of Nagoya University and by the RIKEN Bioresource Center in Japan, respectively.

### 4.7. List of Abbreviations

The list of all abbreviations in this study is shown in Appendix A.

## Figures and Tables

**Figure 1 ijms-24-01523-f001:**
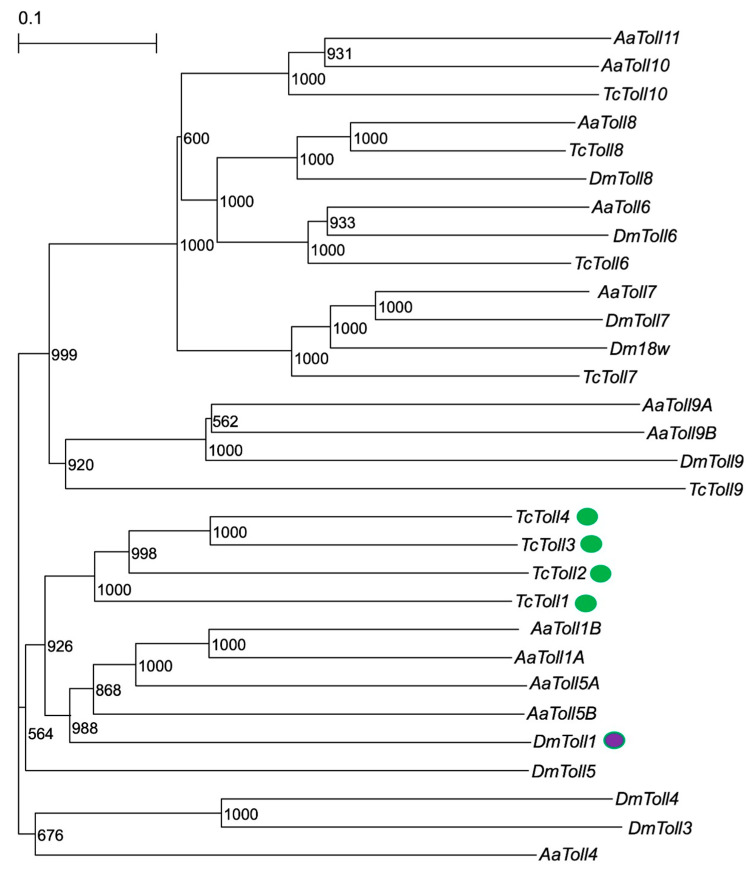
Phylogenetic analysis of *T. castaneum* Toll (TcToll), *D. melanogaster* Toll (DmToll), and *A. aegypti* Toll (AaToll). The accession IDs of the Toll sequences used in the tree are shown in Appendix A. Numbers on branches indicates bootstrap values. Purple and green objects indicate Dm-Toll1 and TcToll1−4, respectively.

**Figure 2 ijms-24-01523-f002:**
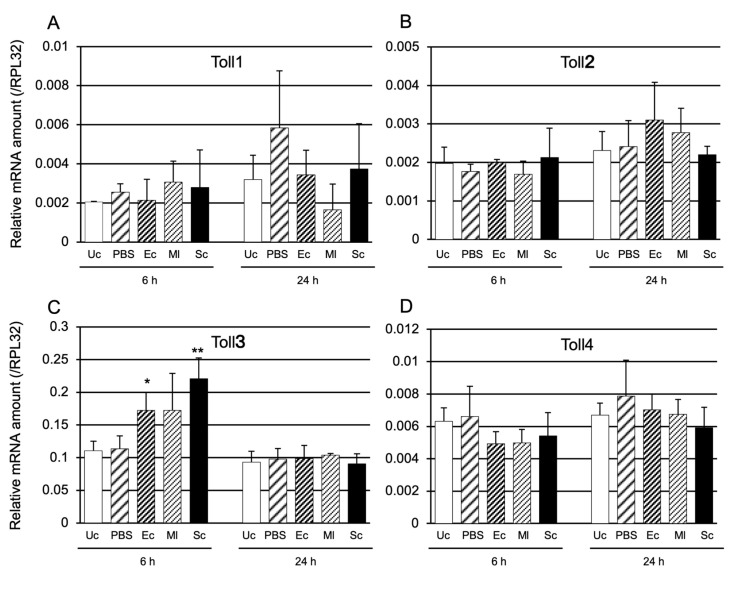
mRNA expression levels of *Toll1* (**A**), *Toll2* (**B**), *Toll3* (**C**), and *Toll4* (**D**) after challenge with three types of microbe. Day-3 pupae were challenged with heat-killed *E. coli* (Ec), *M. luteus* (Ml), or *S. cerevisiae* (Sc). The relative mRNA levels of *Toll1* (**A**), *Toll2* (**B**), *Toll3* (**C**), and *Toll4* (**D**) in the pupae were measured at 6 and 24 h post-injection. Unchallenged (Uc) and phosphate-buffered saline (PBS) injected pupae were used as negative controls. The mRNA amounts of the two genes are shown as relative values to ribosomal protein L32 (*RPL32*) levels in the same samples. Experiments with three animals were independently repeated three times, and each column represents the mean ± S.D. Asterisks and double-asterisks indicate *p* < 0.05 and *p* < 0.01 of Student’s *t*-test compared to Uc samples, respectively.

**Figure 3 ijms-24-01523-f003:**
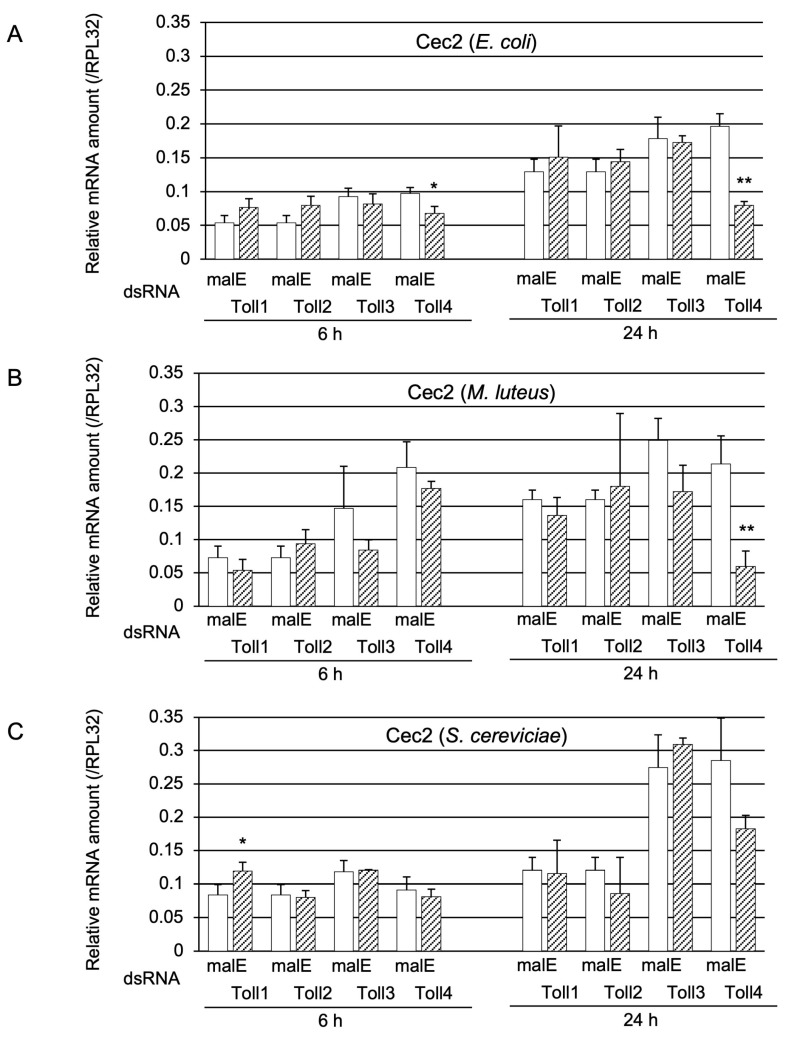
Effect of *Toll* knockdown on *Cec2* induction. mRNA amount of *Cec2* in pupae pre-treated with either malE dsRNA (open bars, control) or *Toll1*, *Toll2*, *Toll3*, or *Toll4* dsRNA (shaded bars) after challenge with either (**A**) *E. coli* (Ec), (**B**) *M. luteus* (Ml), or (**C**) *S. cerevisiae* (Sc). Day-3 pupae were challenged with heat-killed Ec, Ml, or Sc. The relative mRNA levels of *Cec2* in the pupae at 6 or 24 h post injection were measured. The mRNA levels of the genes are shown as relative values to *RPL32* levels in the same samples. Experiments involving three animals were independently repeated three times, and each column represents the mean ± S.D. Asterisks and double-asterisks indicate *p* < 0.05 and *p* < 0.01 compared to Uc samples, respectively, according to Student’s *t*-test.

**Figure 4 ijms-24-01523-f004:**
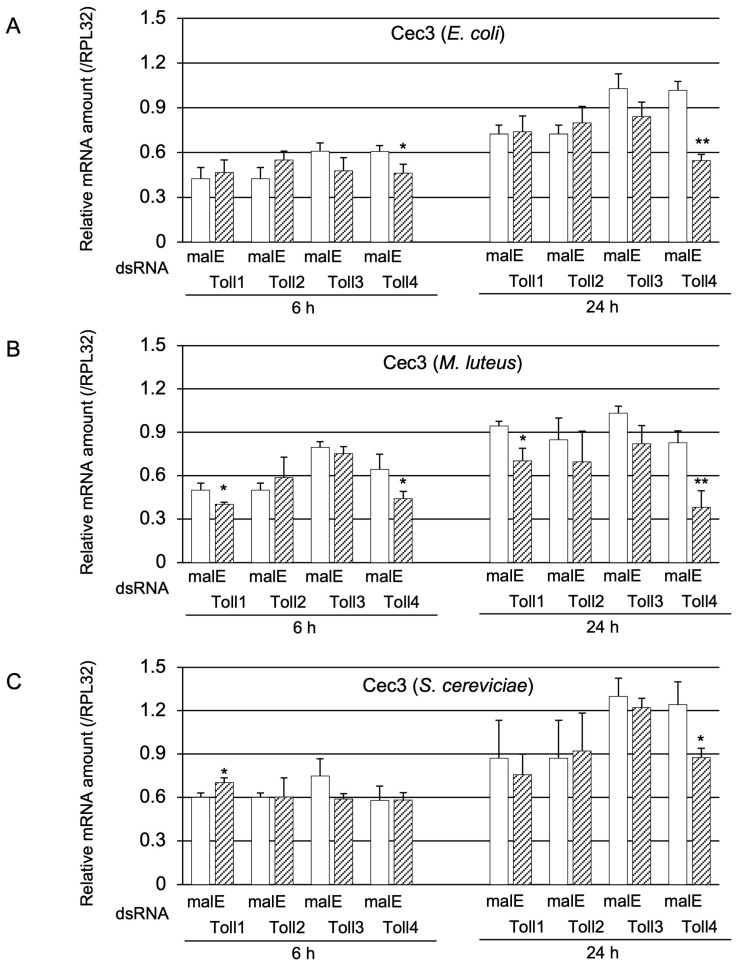
Effect of Toll gene knockdown on *Cec3* induction. mRNA level of *Cec3* in pupae treated with either malE dsRNA (open bars, control) or Toll1, Toll2, Toll3, or Toll4 dsRNA (shaded bars) after challenge with either Ec (**A**), Ml (**B**), or Sc (**C**). The methods of sample preparation and mRNA level measurement were the same as those described in Figure 3. Experiments with three animals were independently repeated three times, and each column represents the mean ± S.D. Asterisks and double-asterisks indicate *p* < 0.05 and *p* < 0.01 compared to Uc samples, respectively, according to Student’s *t*-test.

**Figure 5 ijms-24-01523-f005:**
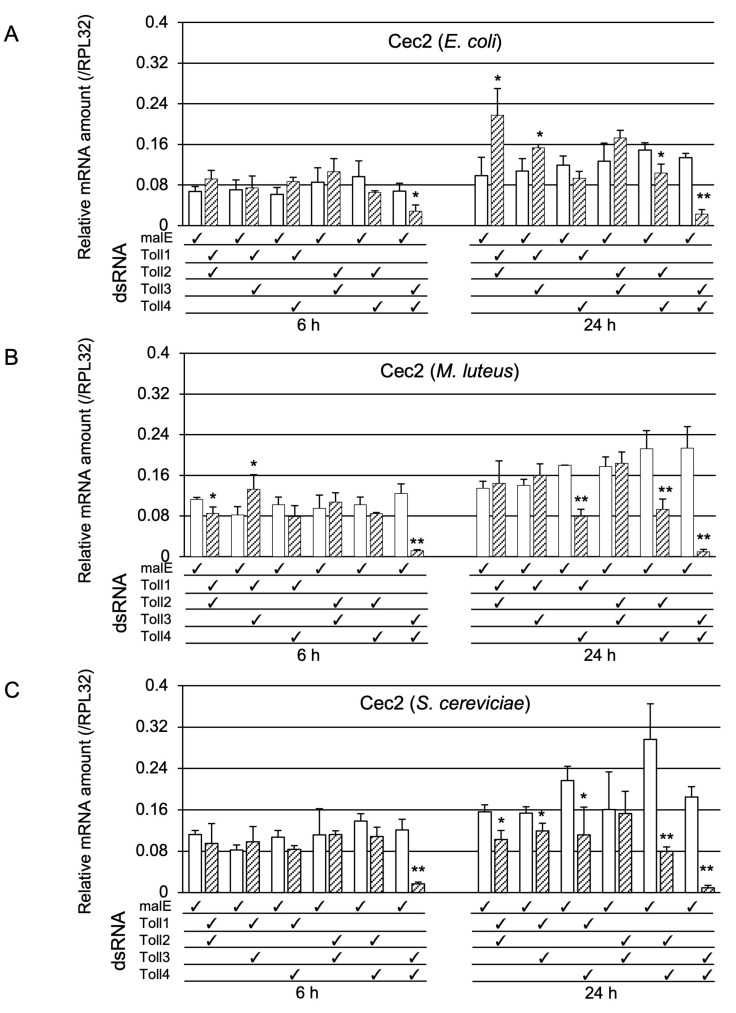
Effect of double knockdown of two Toll genes on *Cec2* induction. mRNA levels of *Cec2* in pupae treated with malE dsRNA (open bars, control) and pupae with double knockdown (check marked) of *Toll1−4* genes (shaded bars) after challenge with either Ec (**A**), Ml (**B**), or Sc (**C**). The methods of sample preparation and mRNA level measurement were the same as those described in Figure 3. Experiments with three animals were independently repeated three times, and each column represents the mean ± S.D. Asterisks and double-asterisks indicate *p* < 0.05 and *p* < 0.01 compared to Uc samples, respectively, according to Student’s *t*-test.

**Figure 6 ijms-24-01523-f006:**
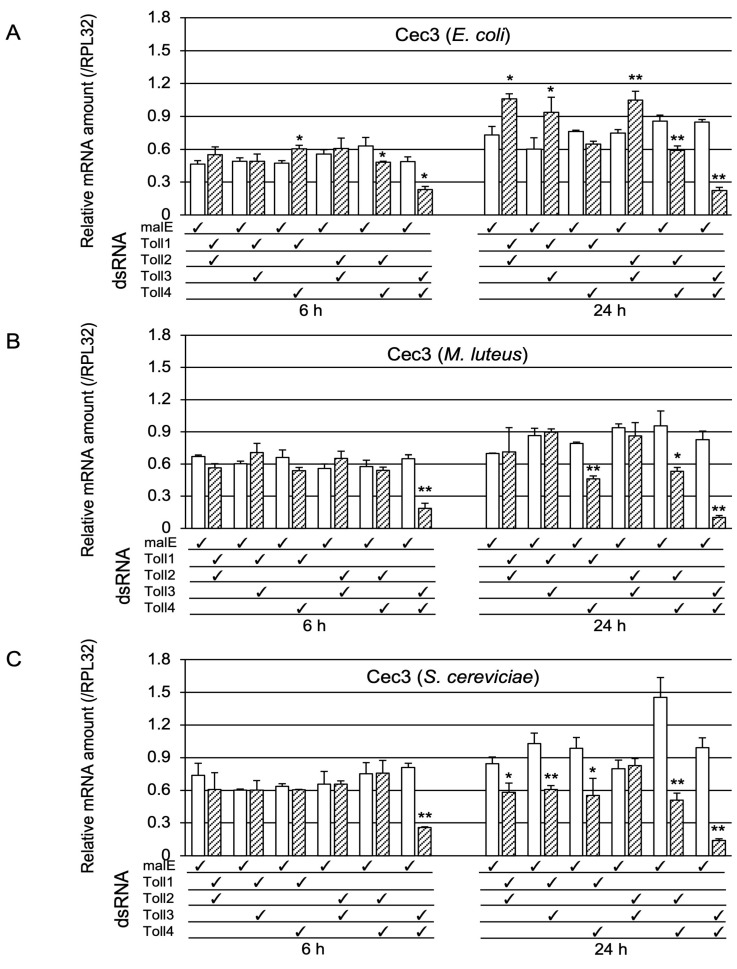
Effect of double knockdown of Toll genes on *Cec3* induction. mRNA levels of *Cec3* in pupae treated with malE dsRNA (open bars, control) and pupae with double knockdown (check marked) of *Toll1−4* genes (shaded bars) after challenge with either Ec (**A**), Ml (**B**), or Sc (**C**). The methods of sample preparation and mRNA level measurement were the same as those described in Figure 3. Experiments with three animals were independently repeated three times, and each column represents the mean ± S.D. Asterisks and double-asterisks indicate *p* < 0.05 and *p* < 0.01 compared to Uc samples, respectively, according to Student’s *t*-test.

**Figure 7 ijms-24-01523-f007:**
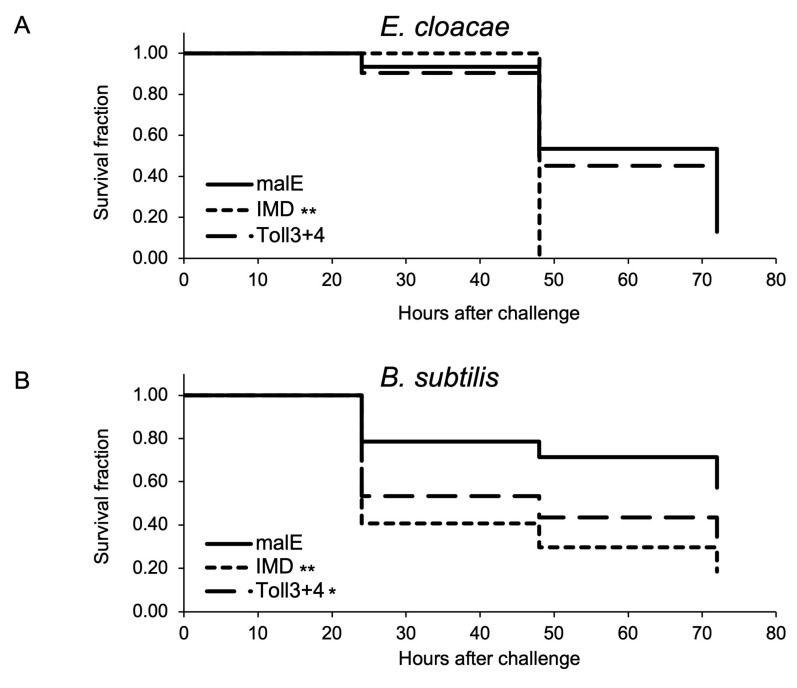
Effects of Toll3 and Toll4 double knockdown on host defenses against two model bacterial pathogens. On day 0, day-3 pupae pre-injected with the respective dsRNAs (*malE* dsRNA-treated pupae and *IMD* dsRNA-treated pupae were used as controls) were challenged with *E. cloacae* (**A**) or *B. subtilis* (**B**), and the survival of the pupae was monitored and recorded every 24 h. Results are shown in Kaplan–Meier plots. Asterisks and double-asterisks indicate *p* < 0.05 and *p* < 0.01, respectively, according to the Gehan–Breslow–Wilcoxon test.

**Figure 8 ijms-24-01523-f008:**
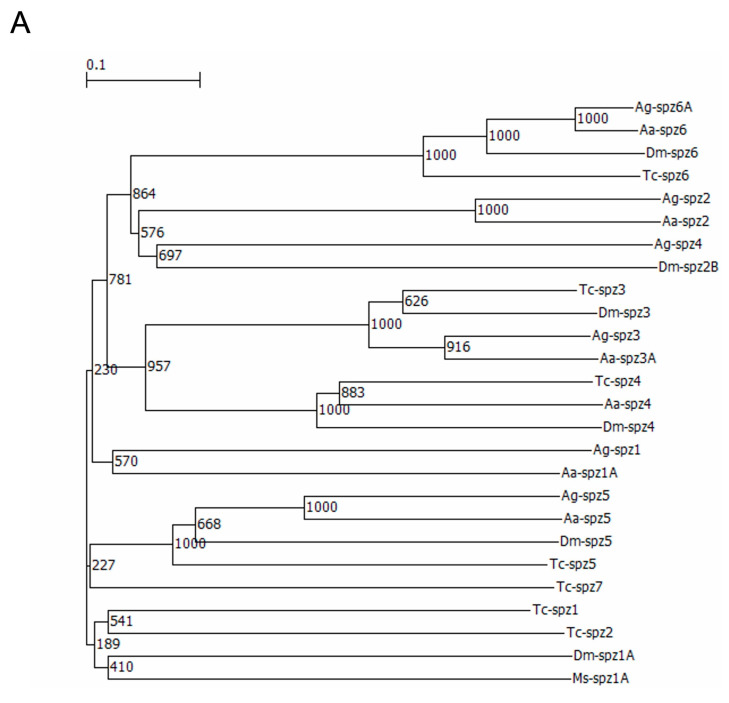
Phylogenetic analysis of spz protein and sequence alignments of spz C terminal sequences. (**A**) A phylogenetic tree was constructed using spz protein sequences of *T. castaneum* (Tc-spz), *D. melanogaster* (Dm-spz), *A. gambiae* (Ag-spz), *A. aegypti* (Aa-spz), and *M. sexta* (Ms-spz) (the accession IDs are listed in Appendix A). Numbers on branches indicate bootstrap values. (**B**) Sequence alignment results of 120 amino acid sequences from spz C-terminal (Appendix A) are shown using BoxShade. White letters in black and gray indicate completely matched or similar residues among the sequences, respectively. Dashed squares indicate proteolytic cleavage sites of arginine residues. The numbers 1, 2, and 3 (bold boxes) above cysteine residues indicate the residues predicted to form the disulfide bond essential for cysteine binding, whereas the number 4 indicates the residue predicted to form dimers of spz proteins.

**Figure 9 ijms-24-01523-f009:**
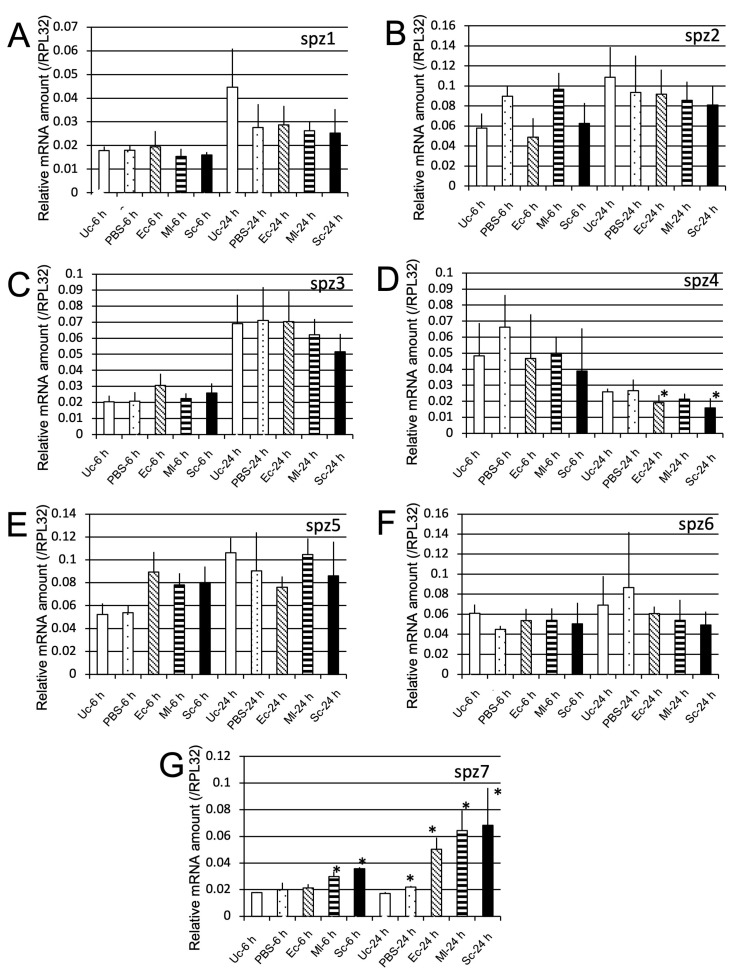
mRNA levels of *spzl1* (**A**), *spz2* (**B**), *spz3* (**C**), *spz4* (**D**), *spz5* (**E**), *spz6* (**F**), and *spz7* (**G**) after challenge with three microbial species. The mRNA levels of the genes are shown as relative values to RPL32 mRNA levels in the same samples. Experiments with three animals were independently repeated three times, and each column represents the mean ± S.D. Asterisks indicate *p* < 0.05 compared to Uc samples, according to Student’s *t*-test.

**Figure 10 ijms-24-01523-f010:**
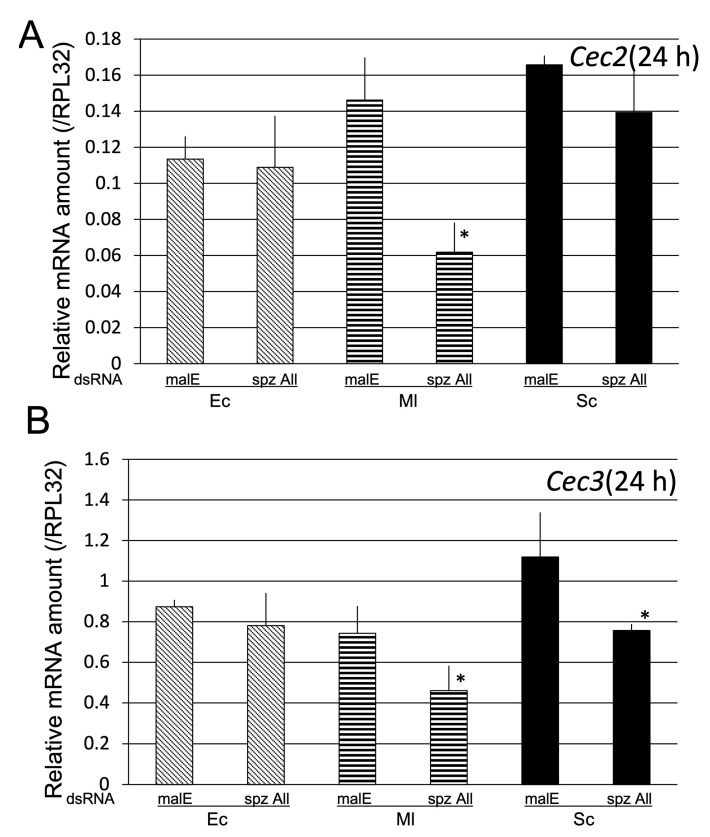
Effect of knockdown of all seven spz genes on *Cec2* (**A**) and *Cec3* (**B**) induction by Ec, Ml and, Sc. mRNA levels of *Cec2* (**A**) and *Cec3* (**B**) in pupae pretreated with malE dsRNA (control) and all seven *spz* dsRNAs (spz All) at 24 h post-challenge with Ec, Ml, or Sc. The mRNA levels of the genes are shown as relative values to *RPL32* mRNA levels in the same samples. Experiments with three animals were independently repeated three times, and each column represents the mean ± S.D. Asterisks indicate *p* < 0.05 compared to malE dsRNA-treated control samples, according to Student’s *t*-test.

**Figure 11 ijms-24-01523-f011:**
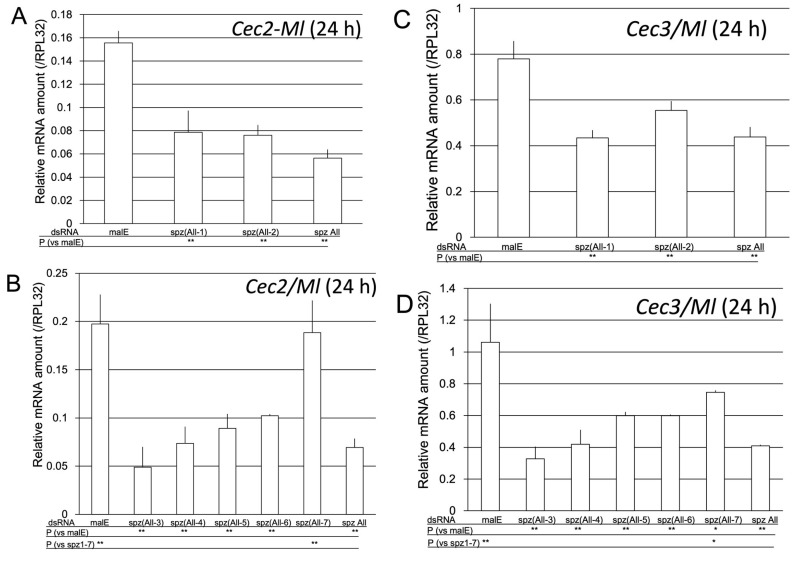
Effect of knockdown of six spz genes on *Cec2* (**A**) and *Cec3* (**B**) induction by *M. luteus* (Ml). The mRNA levels of *Cec2* (**A**,**B**) and *Cec3* (**C**,**D**) were measured in pupae pretreated with malE dsRNA (control), all seven *spz* dsRNAs (spz All, control), or six spz dsRNAs (e.g., spz(All-3) indicates knockdown of *spz1, 2, 4, 5, 6*, and *7*) at 24 h after Ml challenge. The mRNA levels of the genes are shown as relative values to *RPL32* mRNA levels in the same samples. Experiments with three animals were independently repeated three times, and each column represents the mean ± S.D. Asterisks and double-asterisks indicate *p* < 0.05 and *p* < 0.01 (simultaneous confidence interval) compared to malE dsRNA-treated samples (P (vs. malE)) or spzAll samples (P (vs. spz1–7)), respectively, according to Tukey’s honestly significant difference test. The P (vs. spz1–7) row was omitted in (**A**,**C**) because the *Cec2* or *Cec3* mRNA level in spz(All-1) and spz(All-2) pupae was not significantly different from that in spz All pupae.

**Figure 12 ijms-24-01523-f012:**
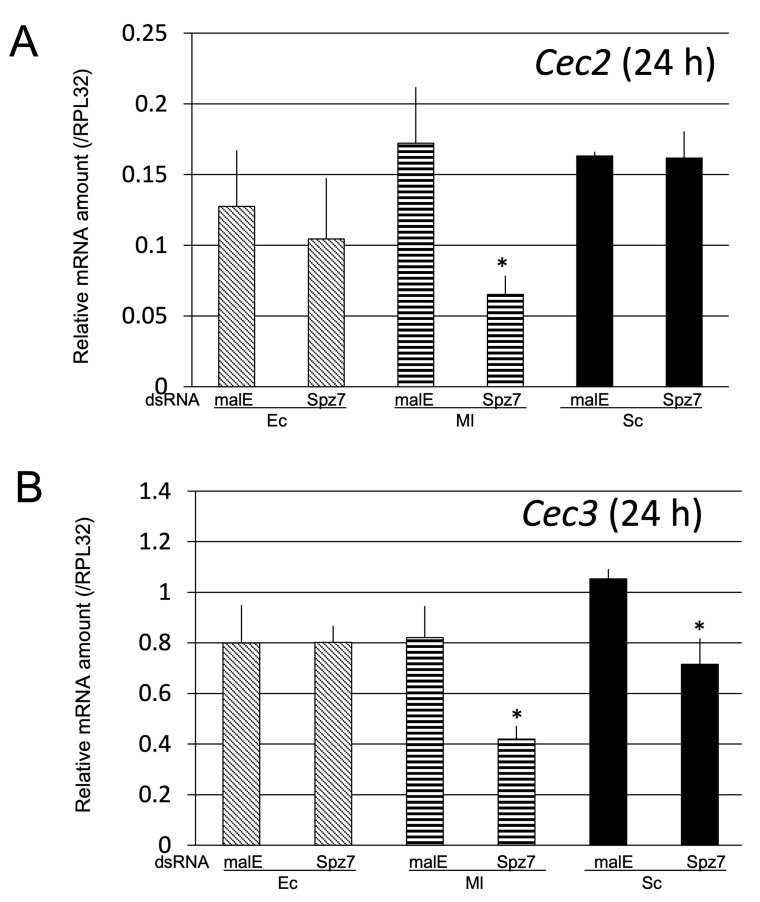
Effect of *Spz7* knockdown on *Cec2* (**A**) and *Cec3* (**B**) induction at 24 h post-challenge. The mRNA levels of *Cec2* (**A**) and *Cec3* (**B**) in spz7 knockdown or control (*malE* dsRNA-treated) pupae were measured. The detailed procedure was the same as that described in Figure 10. Experiments with three animals were independently repeated three times, and each column represents the mean ± S.D. Asterisks indicate *p* < 0.05 compared to malE dsRNA-treated control samples, according to Student’s *t*-test.

**Figure 13 ijms-24-01523-f013:**
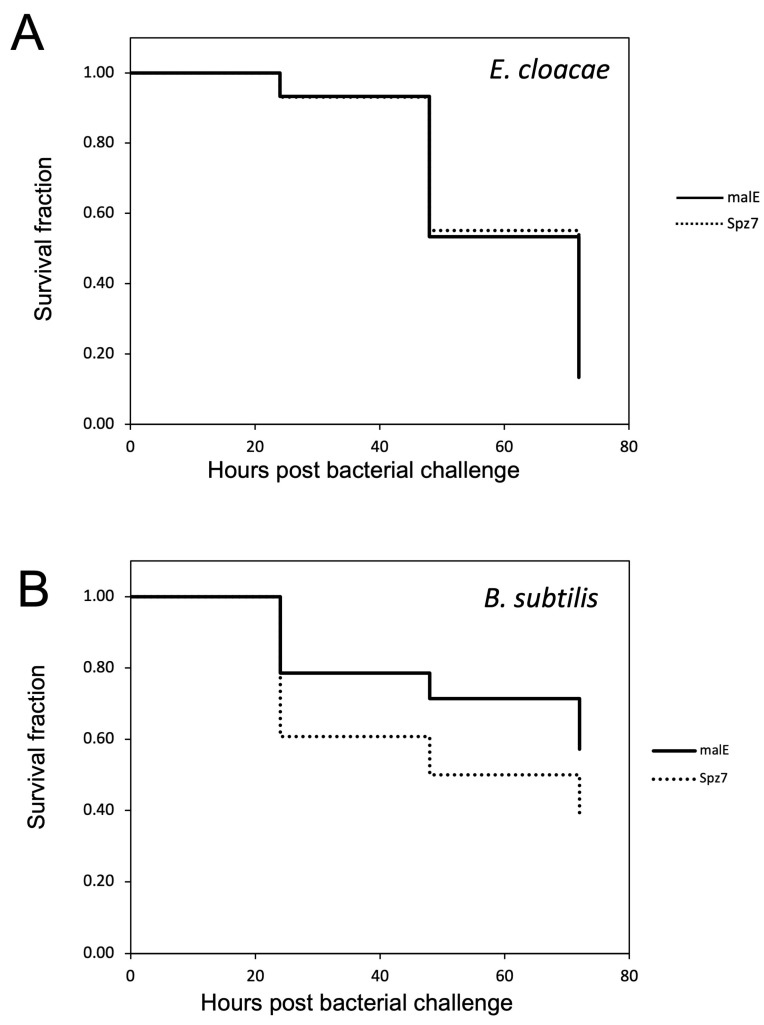
Effects of *spz7* knockdown on host defense against two model bacterial pathogens, *E. cloacae* (**A**) or *B. subtilis* (**B**). The survival of pupae was monitored and recorded every 24 h. The procedure of the assay was the same as that presented in Figure 7. There was no significant difference in survival between control (malE dsRNA-treated) and *spz7* knockdown pupae (*p* > 0.05, according to the Gehan–Breslow–Wilcoxon test).

## Data Availability

All data in this study are available in figshare as described in “Appendix A”.

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
