# Peer review of "Analysis of the Toll and Spaetzle Genes Involved in Toll Pathway-Dependent Antimicrobial Gene Induction in the Red Flour Beetle, Tribolium castaneum (Coleoptera; Tenebrionidae)"

_ijms, 2023, doi:10.3390/ijms24021523_

Round 1

Reviewer 1 Report

The manuscript entitled "Analysis of the Toll and spaetzle genes involved in Toll path- way-dependent antimicrobial gene induction in the red flour
beetle, Tribolium castaneum (Coleoptera: Tenebriodae)" describes the innate immunity of red flour beetle to certain bacterial and fungal challengers. The  manuscript is very well written and presented in scientific way. There are certain spellings and formatting issues e.g., the scientific names at most of the places in references is not italic including some more minor issues. All issues have been highlighted in the text for improvement

Reviewer 2 Report

The production of antimicrobial peptides (AMP) is the main immune reaction in insects. It is already well known that in many insects, the reaction is regulated by the Toll and immune deficiency (IMD) pathways. Authors here continued their studies on the red flour beetle to show that revealed that among nine Toll genes and seven spz genes in the beetle, Toll3 and Toll4, and spz7 are involved in the Toll-dependent AMP gene induction by microbial challenge.

The experiments were carefully done, and the manuscript is professionally written. Nevertheless, it is not easy to read the manuscript because of the abundance of complex figures. I would like to see the results presented more clearly. Authors should also present a complete list of abbreviations used in the manuscript. A careful check of misspellings is also necessary (e.g., students or Students t-test, species names in italics, use of lowercase and uppercase letters, etc.).  Figure legends should explain only those p-values shown in the respective figure.
